# Canine bocavirus-2 infection and its possible association with encephalopathy in domestic dogs

Chutchai Piewbang[1,2], Sabrina Wahyu Wardhani[2,3], Wichan Dankaona[1,4], Sitthichok Lacharoje[1], Poowadon Chai-in[4], Jakarwan Yostawonkul[3,4], Jira Chanseanroj[5], Suwimon Boonrungsiman[4], Tanit Kasantikul[6], Yong Poovorawan[5], Somporn Techangamsuwan[1] *

1 Faculty of Veterinary Science, Department of Pathology, Chulalongkorn University, Bangkok, Thailand, 2 Faculty of Veterinary Science, Animal Virome and Diagnostic Development Research Group, Chulalongkorn University, Bangkok, Thailand, 3 Faculty of Veterinary Science, The International Graduate course of Veterinary Science and Technology (VST), Chulalongkorn University, Bangkok, Thailand, 4 National Nanotechnology Center (NANOTEC), National Science and Technology Development Agency (NSTDA), Thailand Science Park, Pathumthani, Thailand, 5 Faculty of Medicine, Center of Excellence in Clinical Virology, Chulalongkorn University, Bangkok, Thailand, 6 Clemson Veterinary Diagnostic Center, Clemson University, Columbia, South Carolina, United States of America

* somporn62@hotmail.com

**Data Availability Statement:** All the data supporting our findings are contained within the manuscript. Six full-length coding CBoV-2 sequences have been deposited in NCBI GenBank

## Abstract

Canine bocaviruses (CBoVs) have been recognized as pathogens associated with intestinal diseases. Hematogenous spreading caused by CBoV has been documented and may potentiate the virus entry across the blood-brain barrier to initiate a brain infection. This study focused attention on CBoV detection in cases of encepahlopathy and attempted to determine its viral localization. A total of 107 dog brains that histologically exhibited encephalopathy (ED) were investigated for the presence of CBoVs using polymerase chain reaction (PCR). Thirty-three histologically normal brain samples from dogs were used as a control group (CD). CBoV-2 was detected in 15 ED dogs (14.02%) but not in CD dogs ($p = 0.02$), while no CBoV-1 and -3 were detected. Among the CBoV-2 positive dogs, brain histological changes were characterized by nonsuppurative encephalitis, with inclusion body-like materials in some brains. *In situ* hybridization (ISH) and transmission electron microscopy (TEM) confirmed the presence of CBoV-2 viral particles in glial cells, supporting neurotropism of this virus. ISH signals were also detected in the intestines, lymphoid organs, and the heart, suggesting both enteral and parenteral infections of this virus. Whole genome characterization and evolutionary analysis revealed genetic diversity of CBoV-2 sequences and it was varying among the different countries where the virus was detected. This study points to a possible association of CBoV-2 with encephalopathy in dogs. It also highlights the genetic diversity and cellular tropism of this virus.

under accession MW922648-MW922653. https://www.ncbi.nlm.nih.gov/nuccore/MW922648
https://www.ncbi.nlm.nih.gov/nuccore/MW922649
https://www.ncbi.nlm.nih.gov/nuccore/MW922650
https://www.ncbi.nlm.nih.gov/nuccore/MW922651
https://www.ncbi.nlm.nih.gov/nuccore/MW922652
https://www.ncbi.nlm.nih.gov/nuccore/MW922653.

**Funding:** C.P. was supported by the Ratchadapisek Somphot Fund for Postdoctoral Fellowship, Chulalongkorn University. W.D. received a grant from by The Thailand Research Fund through the Royal Golden Jubilee Ph.D. Program (Grant No. PHD/0021/2561). J.C was supported by The Second Century Fund (C2F), Chulalongkorn University. S.T. is funded by Chulalongkorn University: CU_GR_63_75_31_07 and The Chulalongkorn Academic Advancement Into Its 2nd Century Project, Faculty of Veterinary Science, Chulalongkorn University.

**Competing interests:** The authors have declared that no competing interests exist.

## Introduction

The genus Bocaparvovirus (BoV) belongs to the Parvoviridae family, which contains 25 viral species (1, 2). BoV has been identified in various animals, including numerous primates, lagomorphs, rodents, pinnipeds, chiropterans, ungulates, carnivores, and humans [1,2]. Thus, species classification of BoV viruses (BoVs) is based on the host [2]. Although BoVs are commonly associated with respiratory and intestinal diseases of infected hosts, the pathological roles of these viruses remain unclear [1,3–5].

To date, three bocaviruses that belong to *Carnivore bocaparvovirus* have been discovered in domestic dogs, namely canine bocavirus 1 to 3 (CBoV-1 to -3) [5–8]. CBoV-1, previously named canine minute virus, was first isolated in 1967 (7). Most CBoV-1 infections appear to be asymptomatic, with the exception of those in newborn pups, which can develop a systemic fatal disease [9]. CBoV-2 infection, which was discovered in 2012 in dogs, has recently been considered to be a pathogenic [4]. Previous studies detected CBoV-2 genomes in dogs with intestinal and respiratory diseases [5,10], which showed similarities to intestinal and respiratory diseases linked to bocavirus infections in humans. However, the possible association of CBoV infections with the diseases remains unclear. CBoV-3 was detected as an incidental finding in the liver of a dog in a metagenomic study [6]. Since then, there have been no further reports of CBoV-3 detection. Speculation of CBoV infection associated with intestinal and respiratory diseases has been proposed because of viral localization in related organs in dogs with concurrent otherwise unexplained clinical symptoms [3–5]. The pathogenicity and mode of transmission of CBoVs are unknown due to the lack of an appropriate animal model and an established cell culture system for viral isolation [11].

A number of studies have suggested that various BoVs may serve as etiological agents in diseases with atypical clinical symptoms, including encephalopathy [12–17]. Human bocavirus (HBoV) and viral-like particles have been detected in cerebrospinal fluid derived from patients with encephalitis [16,17]. Studies have also detected HBoV in the central nervous system (CNS) of encephalitic patients with concomitant infections of unknown origin [12,13]. In addition, HBoV has been detected in 3–15% of encephalitis patients, pointing to the possible role of HBoV infection in encephalitis [12]. Furthermore, studies of CBoV in dogs and feline bocavirus (FBoV) in cats showed the presence of these viral genomes in brain tissue using polymerase chain reaction (PCR) [10,18,19]. Although CBoV/FBoV genomes have been detected by PCR assays in CNS-related samples, the virus has not been identified *in situ* in CNS tissue.

Several studies have indicated that BoVs can cause viremia, which may lead to its detection in several organs via hematogenous spread [5,10,12,18]. If BoVs can enter the bloodstream and cause viremia, they may be able to disseminate to other parts of body [11,20–22] and/or localize and cause lesions [11]. As hematogenous spread is the typical route by which viruses enter organs, BoVs may be able to pass through the blood–brain barrier and cause lesions in the CNS [16]. There have been a number of recent reports on BoVs associated with encephalitis [12,13,17]. However, systematic studies on BoV localization in the CNS have not been reported, except for one exception, a single case report of porcine bocavirus (PBoV) infection in a pig with encephalitis [14]. Furthermore, little is known about CBoV cellular tropism and its pathogenic potential in the CNS and other organs. Information on enteral and parenteral infection, replication sites, and possible cellular targets of CBoVs are also lacking.

In this study, we focused on CBoV infection in dogs with clinical features of neurological diseases. We conducted PCR assays to detect CBoV in brain tissue and utilized the *in situ* hybridization (ISH) technique and transmission electron microscopy (TEM) to visualize viral localization in the brains of infected dogs. Our findings also provided information regarding

the genetic diversity and evolutionary patterns of CBoV strains, and CBoV pathogenicity and disease presentation.

## Materials and methods

### Animals, samples, and genomic extraction

Between January 2019 and January 2021, 107 brain samples derived from 107 dogs with clinically suspected encephalopathy (i.e., clinical signs of seizures, behavioral changes, circling, disorientation, weakness, loss of balance, and/or cervical spinal pain) that were submitted for a postmortem examination to the Department of Pathology, Faculty of Veterinary Science, Chulalongkorn University, Bangkok, Thailand, were collected. These samples served as the study group, hereafter referred to as the encephalopathy dog group (ED group). Brain tissues from healthy dogs ($N$ = 33) submitted for a postmortem examination that showed no neuropathological changes served as a control group (CD group). Samples positive for rabies infection via a fluorescent antibody test were excluded. Samples with other anatomical abnormalities, such as hydrocephalus, congenital cerebral or cerebellar hypoplasia, primary or secondary brain tumors, traumatic brain injury, and/or metabolic, toxic, ischemic, and hemorrhagic brain diseases as revealed by histopathology or other laboratory results were also excluded from the study. All experimental protocols were approved by the Chulalongkorn University Animal Care and Use Committee (No. 1631002).

The collected brain samples were aliquoted for routine histology by immersing in 10% neutral buffered formalin (SigmaAldrich, MA, USA). For routine diagnostic virology, they were stored at -80˚C. For the histopathology study, the histology sections were examined by an American-boarded veterinary pathology (TK) for histology description. For virological studies, the brain samples were subjected to viral nucleic acid extraction. Briefly, the brain tissue samples (5 g) were homogenized in 1X phosphate buffered saline using an aseptic technique. The homogenized samples were subjected to viral genomic extraction using a Viral Nucleic Acid Extraction Kit II (GeneAid, Taipei, Taiwan) according to the manufacturer's instructions. The extracted nucleic acids were quantified on the basis of a 260/280 absorbance ratio using a Nanodrop® Lite spectrophotometer (Thermo Fisher Scientific Inc., Waltham, MA, USA). The extracted nucleic acids were then stored at -80˚C. Fresh-frozen and formalin-fixed, paraffin-embedded (FFPE) samples derived from various organs in the ED group were also collected. Data on the ages of the dogs were recorded, and the life stage of each dog was classified according to the guidelines of American Animal Hospital Association [23].

### CBoV detection and genome sequencing

All the extracted brain samples were initially screened for genomic BoV using pan-BoV PCR assays targeting a board range of BoVs [24]. Pan-BoV PCR (panBoV)-positive brain samples were subjected to a pan-canine BoV PCR assay (panCBoV), which can detect CBoV-1 to -3 [25]. The primers, cycling conditions, and protocols were as described in a previous study [5], with minor modifications. Specifically, we used GoTaq Green Master Mix (Promega, Madison, WI, USA) which contains a mixture of Taq DNA polymerase, 400 μM of dNTPs, 3 mM of $MgCl_2$ and PCR buffers, and added 5 μl of extracted nucleic acids to increase detection affinity. The PCR-amplified products were visualized using the QIAxcel capillary electrophoresis platform. The capillary cartridge type and settings used in this analysis have been described previously [26]. The positively amplified fragments were later purified using a NucleoSpin® Extract II kit (Macherey-Nagel, Düren, Germany). The purified fragments were then submitted for bidirectional Sanger sequencing at Macrogen Inc. (Incheon, South Korea) to confirm the presence and genetic diversity of the CBoV genomes. To detect other viruses in these

samples, the CBoV-positive brain samples were screened using an in-house pan-family viro-logical PCR pipeline targeting caliciviruses, coronaviruses, flaviviruses, herpesviruses, para-myxoviruses, parvoviruses [18,26,27], and circoviruses [28]. Moreover, specific PCR assays targeting common viruses associated with neurological diseases in canines, including canine distemper virus, canine adenoviruses, and canine parvovirus [29–31], were also conducted. Based on an initial assessment of genetic diversity from the partial genome sequencing results obtained from the panCBoV PCR assay, we selected six representative CBoV-positive samples for full-length coding genome sequencing and analysis. A primer set used to amplify the com-plete coding genome sequence of the CBoV was designed based on the consensus genomic alignment of previously published CBoV genomes available in the GenBank database. The primers used in this study are listed in S1 Table. The designed primer pair were used in a com-bination of GoTaq Green Master Mix (Promega, Madison, WI, USA) and 5 μl of DNA tem-plate, to amplify sequential CBoV fragments. The cycling conditions included 95˚C, 5 min of initial denaturation, followed by 40 cycles of 95˚C for 30 sec, 50˚C for 1 min, 72˚C for 12 min, with a final extension at 72˚C for 10 min.

## Genetic diversity and evolutionary analysis of the CBoV-positive samples

To investigate the genetic diversity and evolutionary patterns of the CBoVs, six complete cod-ing sequences of the selected positive samples in this study were genetically aligned with previ-ously published CBoV genomes and other BoV genomes derived from various host species (available in the GenBank database) using MAFFT alignment version 7 (http://mafft.cbrc.jp/alignment/server/) and MEGA 7 [32]. Phylogenetic trees were then constructed from these alignments using the maximum likelihood method with HKY+G as a best-fit model of nucleo-tide substitution according to the Bayesian information criterion. All phylogenetic trees were tested with 1,000 bootstrapping replicates. Genome pairwise similarity sequences were calcu-lated using a maximum composite likelihood model. Evolutionary analyses were conducted in MEGA 7. The generated nucleotide identity among the CBoVs is presented in Microsoft Excel format.

To identify the impact of recombination processes on the evolution of CBoV strains, RDP v. Beta 4.94 [33], an integrated, statistical-based recombination detection program, was uti-lized. The settings and criteria used for the analysis of the recombination events were as described previously [26–28,34], with a *p* value of 0.01 used as an acceptable cut-off. Bonfer-roni correction was done as standard. Similarity plot and bootscanning analysis, which are embedded in the SImPlot software package v. 3.5.146 (SCRoftware Baltimore, MD), were then implemented for CBoV strains with recombination events in the RDP. This was done to reconfirm the *in silico* analysis of genetic recombination. The bootscan analysis was tested using the Kimura-2 parameter (K2P) model for 100 replications, with a window size of 200 bp, step size of 20 bp, and transition/transversion ratio of 2, as previously described [18,27,28,34].

To shed light on the evolutionary process of the CBoVs, the alignment of various BoV genomes was used as a template for the evolutionary analysis. Briefly, a data set of 33 complete genome sequences containing 27 CBoV complete genome sequences was retrieved from the GenBank database. This data set was then used in an evolutionary analysis of six potential CBoV strains derived from canine brains in this study. The evolutionary analysis was per-formed using the Bayesian Markov Chain Monte Carlo model, implemented in BEAST v2.4.8 [35]. A jModelTest [36] was performed to identify the best fitting nucleotide substitution model for multiple alignment sequences. The best-fit substitution model under a log-normal relaxed and strict clock model at constant population sizes as priors was implemented to account for varied evolutionary rates among lineages. A coalescent Bayesian skyline tree prior

and empirical base frequencies were conducted under the best-fit clock model and run for 500 million chains, sampling every 50,000th generation, with the first 10% discarded as burn-in. The convergence of parameters was confirmed by calculating the effective sample size using the TRACER program v.1.7.0 [37]. Maximum clade credibility trees were annotated using TreeAnnotator v.1.8.3 [35]. A phylogenetic tree with estimated divergence, variable timeline, posterior probability, and 95% highest posterior density (HPD) was generated and displayed using FigTree v.1.4.3.

## Chromogenic identification of CBoV DNA localization in brain tissues

To support the results of CBoV PCR and hence viral tropism in other organs, seven selected CBoV FFPE tissues in which CBoV PCR was positive were subjected to perform the ISH technique with a chromogenic DNA probe. Briefly, the FFPE tissues were cut at 4 μm, and placed onto a positively charged glass slide. After deparaffinization, rehydration, and proteolytic digestion with 10 ng/ml of proteinase K at 37˚C for 10 min, the FFPE sections were post-fixed and subsequently hybridized with 20 ng/μl of CBoV DNA probe. This probe covered 440 bp of the *NS1* gene and was constructed using a PCR DIG Probe Synthesis Kit (Roche Diagnostics, Basel, Switzerland) according to the manufacturer's recommendations. Following ISH overnight at 50˚C, the reactions were visualized using a combination of 1:200 anti-DIG-AP Fab fragments in blocking solution (Roche Diagnostics, Basel, Switzerland) and Liquid Permanent Red (Dako, Glostrup, Denmark). The sections were then counterstained with hematoxylin. Samples containing red dots marked localization of the virus and were considered positive. Incubation with a feline panleukopenia virus probe [38] rather than the CBoV probe was used as a negative control. A CBoV-2-negative brain section incubated with a CBoV-2 probe was used as an additional negative control.

To confirm CBoV localization in brain tissue, ISH was performed using the same hybridization procedure described earlier, except using a different CBoV probe covering 214 base pairs of the *VP1* gene. StayGreen/AP Plus (Abcam, Cambridge, USA) was used as the detection system. Green precipitates associated with the cells were considered positive. The negative control slides were performed as described above.

## CBoV particles in brain tissues

To support the PCR and ISH evidence of CBoV in brain tissue, virus particles in two CBoV-positive brain sections were analyzed using TEM and a modified pop-off technique, as previously described [27,38]. The sections were postfixed in 1% osmium tetroxide prior to the TEM (HT7800; Hitachi, Tokyo, Japan) study at 80 kV.

## Detection of CBoV localization in other organs

To evaluate cellular tropism of the CBoV, conventional PCR, in combination with ISH targeting the *NS1* gene of CBoV-2, as described in the chromogenic identification of CBoV DNA localization in brain tissues, was used to detect the virus in various organs. Previously known CBoV genomes obtained from a previous study [5] and a no template control were used as positive and negative controls, respectively.

## Statistical analysis

Two-tailed Fisher's exact test was used for statistical anaysis of the various groups, and a *p*-value of < 0.05 was considered to indicate statistical significance. All statistical analyses were performed using GraphPad Prism 8.

## Results

### Genetic screening of CBoV in brain samples

As shown by the initial BoV screening results using a panBoV PCR assay in brain samples, CBoV was detected only in the ED group that revealed 14.02% (15/107) positive samples. There were significant differences in the CBoV-positive dogs that were found only in the ED dogs ($p$ = 0.02). The panBoV-positive PCR samples were then tested using a panCBoV PCR assay to validate the positive results and differentiate the subtypes of CBoV. The results showed that all the panBoV PCR-positive samples were positive in the panCBoV PCR assay. All the samples were identified as CBoV-2, as confirmed by bi-directional sequencing. Among the positive samples, 60% (9/15) were from pups, 26.67% (4/15) were from adult dogs, and 13.33% (2/15) were from senior dogs. All the CBoV-2 positive dogs showed clinical signs of weakness (15/15), and 66.67% (10/15) of CBoV-2 positive dogs had seizures (10/15). Canine circovirus was also detected in one CBoV-positive dog in the ED group.

### Full-length genome analysis of CBoV-2

Six CBoV-positive PCR samples were selected for analysis of the entire genome coding sequences by multiple conventional PCR assays and sequencing. Six 5,000 bp complete CBoV genome coding sequences were obtained: CBoV 036CP1-TH2021, 038CP3-TH2021, 008CP4-TH2021, 240CP5-TH2020, 241CP6-TH2020, and 147CP7-TH2020 (GenBank accession numbers: MW922648-MW922653). Three open reading frames (ORFs) that encoded nonstructural proteins (*NS1* and *NS2*), viral structural proteins (*VP1* and *VP2*), and nucleoprotein (*NP*), respectively, were detected. An additional ORF (4) was identified downstream of the NS1 in the obtained CBoV-2 strain. Partial deletions downstream of the *VP* genes were not observed. Pairwise genetic divergence of the obtained CBoV-2 strain was estimated based on a comparison with previously published CBoV genomes. Of the six CBoV-positive PCR samples, pairwise genetic diversity ranged from 86.4 to 97.0%. However, when compared with a previously detected CBoV-2 strain, Con-161 (JN648103), genetic similarity was lower, ranging from 86.4 to 86.8% (S1 Fig). Genetic recombination analysis revealed no evidence of potential recombination in the obtained CBoV-2 genomes.

### Evolutionary analysis of CBoV

A data set on complete genomes of CBoVs detected in various geographical regions, together with obtained CBoV strains in this study, was used to estimate the divergence timeline and evolutionary history of CBoVs. The general time-reversible (GTR) model with a strict clock model was used as a suitable best-fit substitution model for aligned sequences. A phylogenetic tree of these data revealed four main distinct clades: CBoV-1, CBoV-2 group I (GI), CBoV-2 group II (GII), and CBoV-3 (Fig 1). The overall evolutionary rate was estimated to be $2.36 \times 10^{-4}$ substitutions/site/year (95% HPD: $4.18–5.22 \times 10^{-4}$). The phylogenetic tree based on the complete genome sequences of the six CBoV strains identified in this study showed that they formed two distinct groups, which may have the same origin around 2000. Based on reference strains, five of the six CBoV strains (036CP1, 038CP3, 008CP4, 240CP5, and 241CP6) belong to CBoV-2 GI, which shares the same lineage as an CBoV strain isolated from Hong Kong and South Korea and a BoV strain, 147CP7, belonging to CBoV-2 GII, which exhibits high nucleotide identity to a CBoV strain found in Thailand in 2016 (MG025952).

### Histopathology and CBoV-2 localization in brain sections

In 15 of the CBoV-2 PCR-positive brain samples, histological changes were described collectively, except where otherwise specified. Similar levels of mild-to-moderate degeneration of

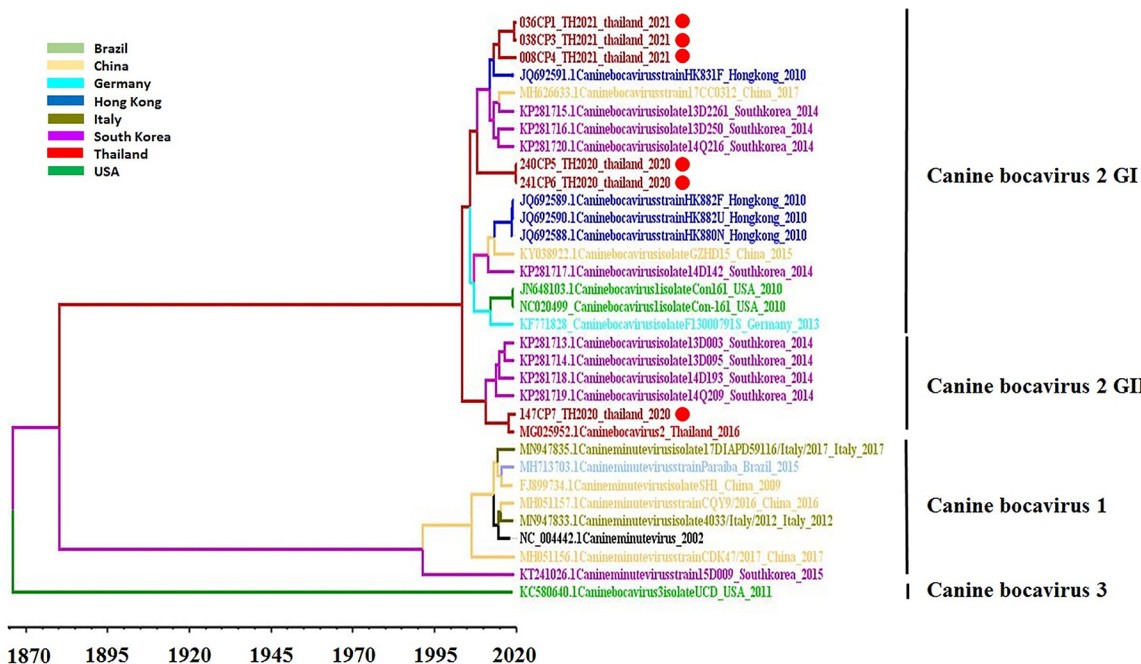

**Fig 1. Maximum clade credibility tree of complete genome sequences of CBoV.** A GTR model with a strict clock model. A coalescent Bayesian skyline tree prior run for 500 million chains. The tree branches are colored according to the sites where the strains were isolated. The red circles represent the CBoV strain collected in this study.

resident cells, including gliosis, darked neurons, and some neuronal vacuolations, were observed in all of 15 PCR-positive brain sections. In the study, 10/15 (66.67%) CBoV-2 PCR-positive brain sections showed minimal-to-mild encephalitis, characterized by nonsuppurative perivascular cuffing. In 4/15 (26.67%) brain sections, hypercellularity of the cortical plate and ventricular zone of the cerebrum was observed, with dense aggregates of glial cells, neurons, and neuropils, some of which were edematous. Some neurons were triangular and hypereosinophilic, with nuclear pyknosis (neuronal necrosis). In two of four brain specimens examined, variable numbers of glial cells containing large, bright eosinophilic intranuclear inclusion-like material were observed. One of two specimens with inclusion-like material had diffuse thickening of meninges, with markedly engorged blood vessels and increased prominence of spindloid stromal cells. Mild interstitial and perivascular infiltration of lymphocytes and plasma cells were also observed in this specimen. In the cerebellum, meningeal blood vessels in the sulci and gyri were markedly engorged (Fig 2A).

Viral localization of the selected CBoV-2 PCR-positive brain samples in the CNS was investigated using two independent ISH techniques with different targeting probes. Six of seven brain sections (85.71%) tested positive using both ISH methods, with five of seven sections (71.42%) showing strong nuclear signals in many glial cells, which were located at the cortical plate and, to lesser extent, within the ventricular zone of the cerebrum (Fig 2B–2D). Nuclear positive signals were strong where inclusion body-like materials were observed. The results of the two independent ISH methods were similar: There were no signals detected in the cerebellum and the brain stem of the examined sections, and no hybridization signals were observed in the negative control section using a feline parvovirus probe (S2 Fig).

To confirm the results of the two independent ISH procedures and the presence of CBoV-2 particles in brain tissues of the infected dogs, two selected FFPE-brain sections with viral-like inclusion bodies in glial cells that were positive in both ISH assays were subjected to TEM.

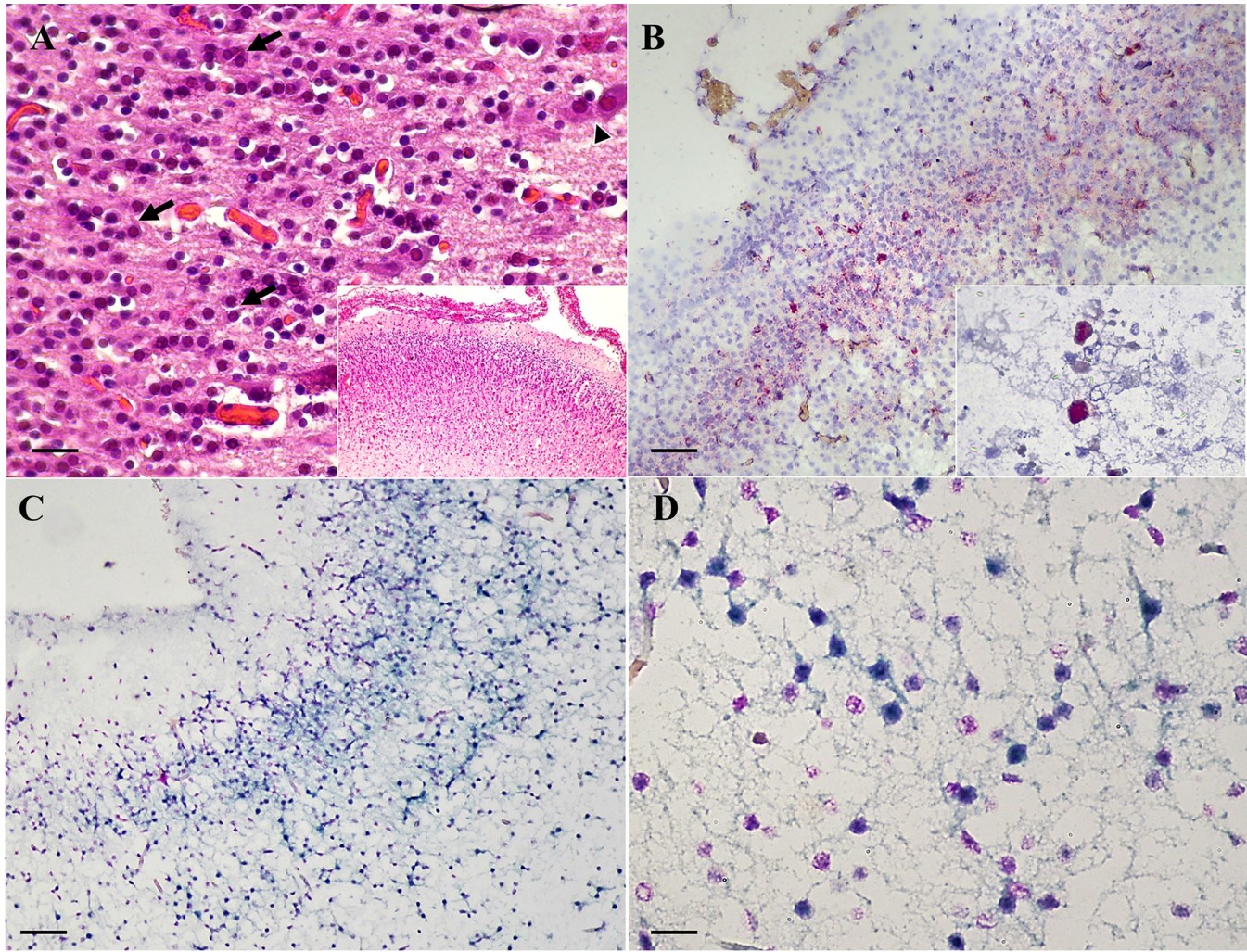

**Fig 2. CBoV-2 infection in dog no. 240CP5.** Photomicrographs demonstrate H&E staining **(A)** and CBoV-2 ISH **(B–D)** in brain. (A). Hypercellularity of the cortical plate and ventricular zone of the cerebrum (inset). Variable numbers of glial cells containing large, bright eosinophilic intranuclear inclusion-like material (arrows) and hypereosinophilic neurons (arrowhead). (B). Hybridization signals using a probe targeting *NS1* of CBoV-2 were strong and localized in the nucleus of glial cells (red precipitates). The signals were strong and clearly observed in the glial cells containing intranuclear inclusion-like material (inset). Hematoxylin (pale blue) was used as a counterstain. (C) and (D). Nuclear hybridization signals using a probe targeting *VP1* of CBoV-2 revealed diffuse staining patterns in many glial cells (dark blue green). Nuclear fast red (pink) was used as a counterstain. Bars indicate 25 μm (A and D) and 120 μm (B and D).

TEM revealed the ultrastructural localization of CBoV-2, with numerous electron-dense icosahedral viral particles, measuring about 20 nm in diameter, aggregated within large, dense intranuclear inclusion elements within the glial cells (Fig 3).

## Enteral and parenteral localization of CBoV-2

To explore the potential distribution of CBoV-2 in different organs, both conventional PCR and ISH methods were used. The PCR assay was conducted to detect CBoV-2 in various organs, including the intestines and brains of all the tested dogs. Lymph nodes, thymus, kidneys, heart, brain, spleen, and trachea were differentially positive among the cases (S2 Table).

To elucidate the presence of CBoV-2 genes in various organs due to either hemorrhagic spreading or actual localization as viral tropism, ISH was performed in samples that were PCR positive. Apart from the brain, strong nuclear hybridization signals were detected in villous

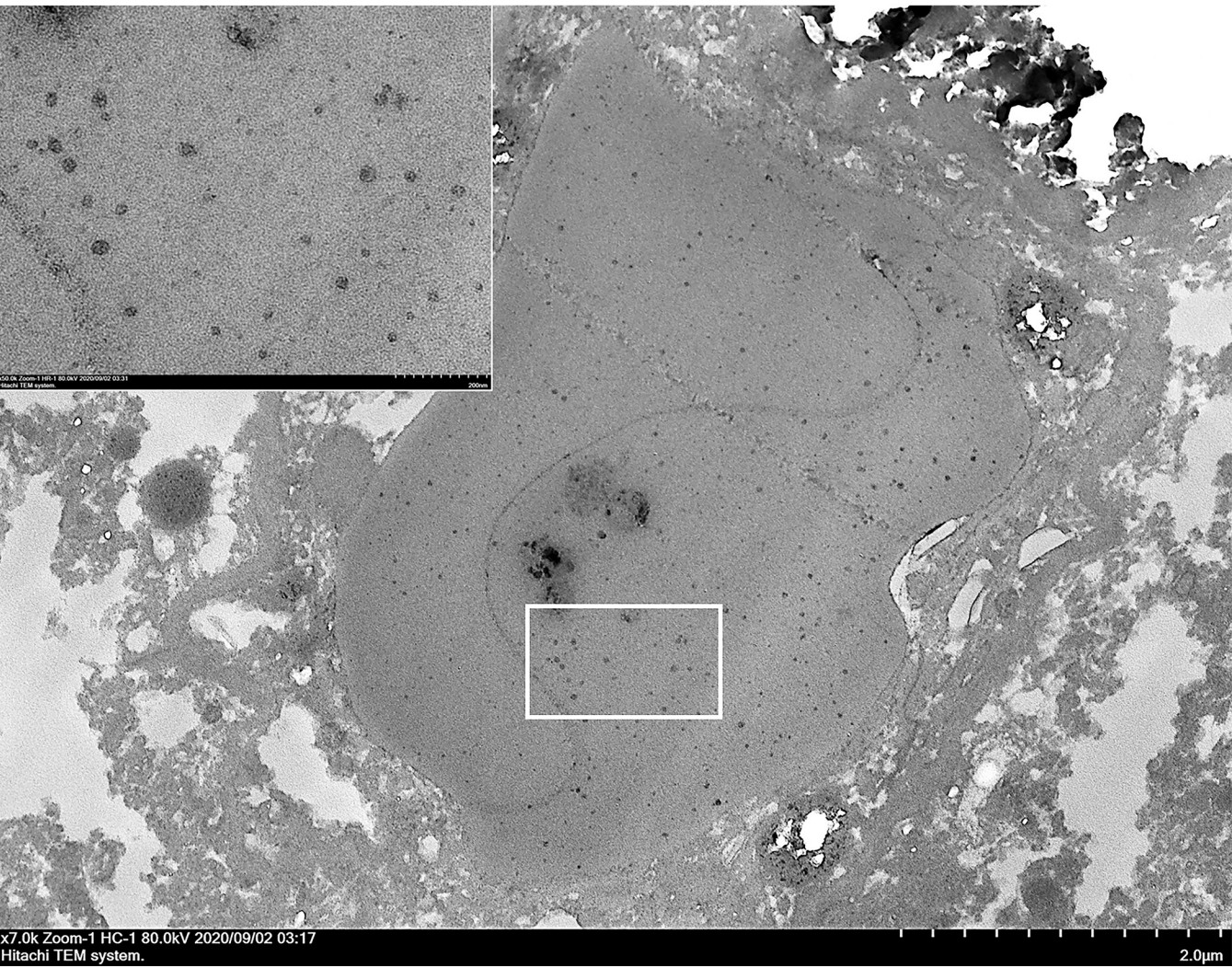

**Fig 3. CBoV-2 infection in dog no. 241CP6, showing the ultrastructure of an infected brain.** An oligodendroglia cell containing an inclusion body within the nucleus, which contained aggregates of electron-dense icosahedral virions (inset). Bar as indicated.

epithelia and lymphocytes in thymus and lymph nodes (Fig 4). Weaker ISH-positive signals were noted in the nucleus of the myocardium (Fig 4E). No signals were detected in other tested organ sections. Details on the PCR and ISH results are provided in S2 Table.

## Discussion

Viremia induced by BoVs increases the possibility of dissemination of the virus to various organs and tissues [5,11,18,21,22]. If the virus can adapt to infect resident cells, novel tropism may result in infection of new cells. Apart from gastrointestinal and respiratory diseases, recent studies have implicated HBoVs in encephalopathy/encephalitis [12,13,15–17]. In this study, we hypothesized that CBoVs may disseminate to the CNS where they may be associated with neurological disorders in canines, similar to those found in HBoV-infected humans. However, this hypothesis is not definitively proved by fulfilling the Koch's postulates due to the lack of a cell culture system for CBoV-2 propagation. Therefore, this study only conducted the controlled study of diseased and healthy animals to provide insight into this assumption.

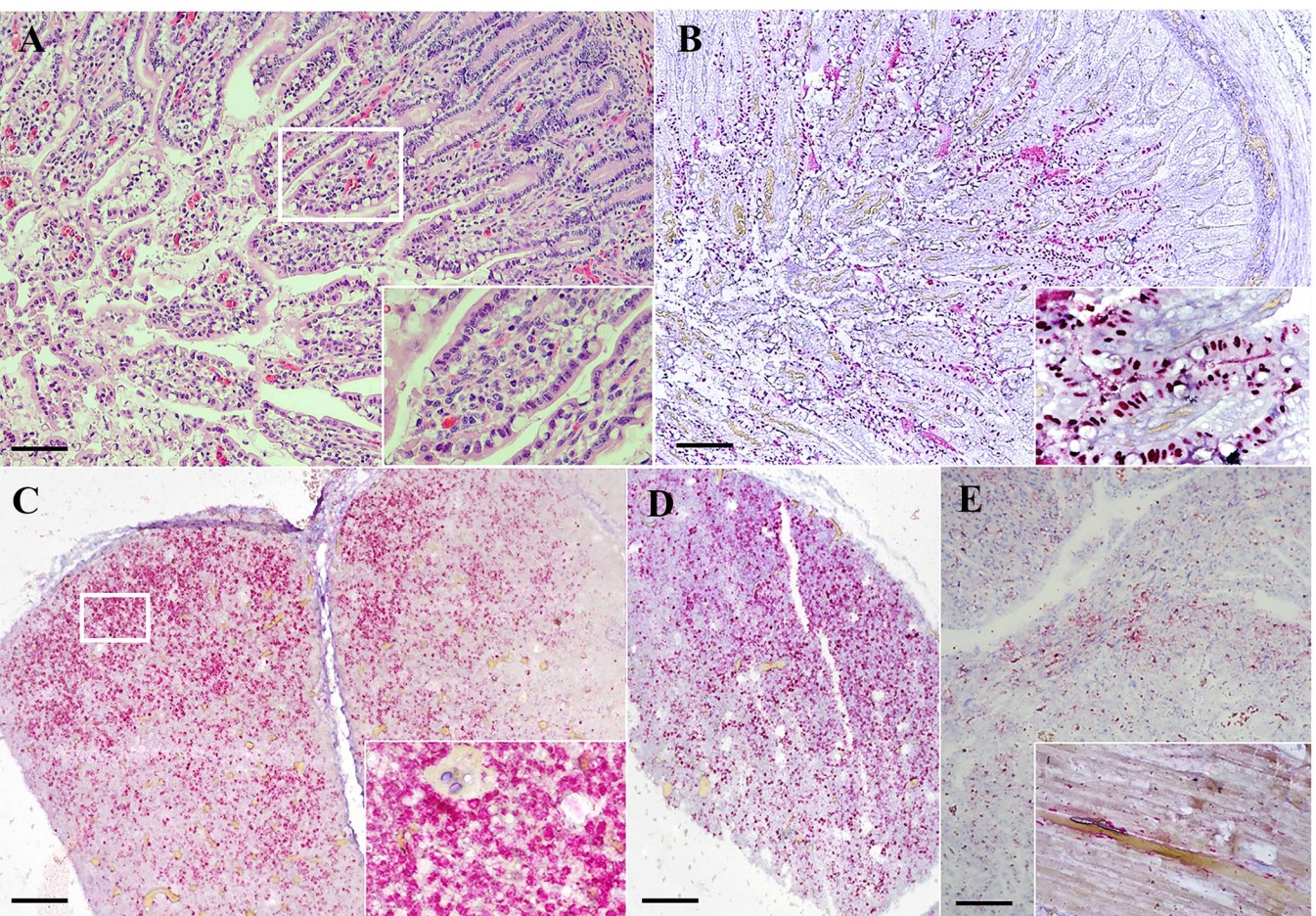

**Fig 4.** CBoV-2 infection in dogs no. 008CP4 (A-B) and no. 038CP3 (C-E). Photomicrographs demonstrate H&E staining (A) and CBoV-2 ISH (B–E). Variable numbers of enterocytes contained large, bright eosinophilic, intranuclear inclusion bodies (inset). (B). Diffuse strong hybridization signals were localized in the nucleus of enterocytes (red precipitates), the cells of which contained inclusion bodies (inset). (C) Strong hybridization signals were observed in the nucleus of mononuclear cells (inset) of the thymus and (D) mesenteric lymph nodes. (E) Nuclear hybridization signals were observed in myocardial cells (inset). Hematoxylin (pale blue) was used as a counterstain. Bars indicate 25 µm (A–E).

Since this study investigated on the rabies-negative brain samples, we only detected the CBoV-2 genome (with no presence of other common canine viral neuropathogens such as CDV, CAdVs, and CPV) in brain tissue of the ED group but not the CD group. The detection of CBoV-2 in ED brains corresponded with optical determination of viral localization using ISH and TEM, suggesting that CBoV-2 plays a potential role in encephalitis. This study also revealed the presence of CBoV-2 antigen in other organs, corresponding with the PCR and ISH results.

Limited data point to *Carnivore bocavirus* in the CNS [10,18,19]. However, several studies have pointed to the potential role of HBoVs in encephalitis [12,13,15–17]. PBoV infection without concomitant infections was reported in a pig with signs of encephalitis, and this study confirmed neurotropism of the PBoV [14]. Given the current evidence for BoV neurotropism, in this study, we focused on CBoV detection in brain samples obtained from dogs with signs of encephalopathy. We detected CBoV-2 in brain tissues in the ED group but no CBoV genotypes in the CD group. Thus, CBoV-2 presumably plays a dominant role in canine encephalopathy. As the sample size in our study was small, the findings may not truly reflect the

epidemiology of CBoVs. Further large-scale retrospective and perspective studies are needed to elucidate the incidence of CBoV genotypes in dogs with encephalopathy.

Intestinal epithelia and lymphoid cells have been recognized as sites of cellular tropism in most *Carnivore bocavirus*, including CBoV-1 and CBoV-2 [5,7,10,39] and FBoV-1 [18]. In the present study, CBoV-2 was localized in intestinal epithelia and lymphoid tissues, including lymph node and thymus, supporting the idea of cellular tropism. As noted earlier, carnivore bocaviral DNA has been found in brain tissues via PCR methods (10, 18). However, no studies have confirmed virus localization in the brain. Thus, the present study demonstrates a potentially novel cellular tropism of CBoV-2 in brain.

In addition, we detected CBoV-2 in myocardium, which suggests that the heart may serve as another target organ for CBoV-2 infections. Note also that CBoV-1, a counterpart of CBoV-2, is known to be associated with myocarditis [40]. In the present study, although we detected CBoV-2 DNA in all the selected brain tissue samples, as well as in several other organs in the infected dogs, not all brain samples or PCR-positive tissues revealed ISH signals to support the presence of viral DNA. Hematogenous spreading of the virus or low amounts of virus present in the tested organs may be hypothesized for this finding. However, we did not investigate the presence of CBoV-2 in blood of these cases, further investigations are needed to support this hypothesis. Evidence of viral hematogenous spreading of CBoV-2 and other BoVs has been reported, in addition to viral gene/genome detection in various organs [5,18,21,22]. The incongruence between our PCR and ISH data may be caused by a difference in sensitivity, as ISH has lower sensitivity than PCR [41,42]. In this study, we postulated that if CBoV-2 causes viremia, it could disseminate to various organs in the body and infect resident cells. Future studies using animal models or primary culture systems susceptible to CBoV-2 infection are needed to shed light on this issue.

Several recent studies have shed light on the genetic diversity of CBoV-2 [3,4,10,24,25,43]. Although CBoV-2 is a DNA virus, which has a low potential to mutate, recent studies indicated that CBoV-2 has a major dynamic range of its evolution due to natural genetic recombination [5,44]. Previous studies reported partial deletion and the presence of a unique ORF, ORF4, in the *VP* gene of CBoV-2, with the authors speculating that this ORF may play a role in CBoV-2 disease presentation and pathogenicity [3–5]. These observations required us to investigate the genetic diversity of CBoV-2 in the present study. Although a previous study described CBoV-2 genomes derived from dogs in Thailand provided evidence of genetic recombination [5], we found no evidence of genetic recombination in these obtained CBoV-2 genomes in the present study. However, we found that the obtained CBoV-2 genomes were diverse and disperse. The evolutionary data obtained in this study revealed that CBoV subgroup diversity can occur frequently. This finding is supported by molecular epidemiology studies of CBoV strains isolated in different districts and countries [5,10,24,43]. Intensive monitoring and tracking of genetic diversity and evolutionary forces of CBoV-2 are needed in further research.

## Conclusions

This study provides novel insights into CBoV-2 tropism, suggesting that CBoV-2 plays a potential role in canine encephalopathy. Our data on viral localization indicate that CBoV-2 may also infect lymphoid and myocardial cells. Furthermore, we hypothesize the hematogenous spreading of bocaviruses. Hematogenous spreading and viral dissemination may result in different disease presentations. Finally, this study provides novel avenues for future research on CBoV-2, and the CBoV-2 infection in dogs may serve as a model for comparative studies of other BoV infections.

## Supporting information

**S1 Fig. Pairwise nucleotide identity of obtained CBoV-2.** The highest and lowest nucleotide identity of obtained CBoV-2 are highlighted with red and yellow colors, respectively.
(TIF)

**S2 Fig. CBoV-2 infection.** Photomicrographs of dog no. 240CP5. *In situ* hybridization with feline panleukopenia probe incubation served as negative controls of (A) brain, (B) intestine, (C) thymus, and (D) heart sections. Bars indicate 25 μm for (A) and 120 μm for (B-D).
(TIF)

**S1 Table. The CBoV-2-specific primers used for the full-length genome sequencing and for *in situ* hybridization (ISH).** The primers were designed based on alignment of multiple CBoV-2 genomes available in GenBank.
(DOCX)

**S2 Table. Detection of CBoV-2 in various organs of selected dogs.** The CBoV-2 genomes were detected using conventional polymerase chain reaction (PCR) and *in situ* hybridization (ISH).
(DOCX)

## Author Contributions

**Conceptualization:** Chutchai Piewbang, Tanit Kasantikul, Somporn Techangamsuwan.

**Data curation:** Chutchai Piewbang, Jakarwan Yostawonkul.

**Formal analysis:** Chutchai Piewbang, Sabrina Wahyu Wardhani, Wichan Dankaona, Jira Chanseanroj.

**Funding acquisition:** Somporn Techangamsuwan.

**Investigation:** Chutchai Piewbang, Sabrina Wahyu Wardhani, Sitthichok Lacharoje, Jakarwan Yostawonkul, Tanit Kasantikul.

**Methodology:** Chutchai Piewbang, Poowadon Chai-in, Tanit Kasantikul.

**Project administration:** Chutchai Piewbang, Suwimon Boonrungsiman, Tanit Kasantikul, Somporn Techangamsuwan.

**Resources:** Poowadon Chai-in, Jira Chanseanroj, Suwimon Boonrungsiman, Yong Poovorawan.

**Software:** Jira Chanseanroj.

**Supervision:** Suwimon Boonrungsiman, Tanit Kasantikul, Yong Poovorawan, Somporn Techangamsuwan.

**Validation:** Chutchai Piewbang.

**Writing – original draft:** Chutchai Piewbang.

**Writing – review & editing:** Chutchai Piewbang, Somporn Techangamsuwan.

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
