## [Decision Letter · Decision Letter 0]

2 Jul 2021

PONE-D-21-17173

Canine Bocavirus-2 and its possible association with encephalopathy in domestic dogs

PLOS ONE

Dear Dr. Piewbang,

Thank you for submitting your manuscript to PLOS ONE. After careful consideration, we feel that it has merit but does not fully meet PLOS ONE’s publication criteria as it currently stands. Therefore, we invite you to submit a revised version of the manuscript that addresses the points raised during the review process.

Many thanks for submitting your manuscript to PLOS One

It was reviewed by two experts in the field, and they have recommended some modifications be made prior to acceptance

I therefore invite you to make these changes and to write a response to reviewers which will expedite revision upon resubmission

I wish you the best of luck with your modifications

Hope you are keeping safe and well in these difficult times

Thanks

Simon

We look forward to receiving your revised manuscript.

Kind regards,

Simon Clegg, PhD

Academic Editor

PLOS ONE

3. We note that Figures 2, 3, 4 and S2 in your submission contain copyrighted images. All PLOS content is published under the Creative Commons Attribution License (CC BY 4.0), which means that the manuscript, images, and Supporting Information files will be freely available online, and any third party is permitted to access, download, copy, distribute, and use these materials in any way, even commercially, with proper attribution. For more information, see our copyright guidelines: http://journals.plos.org/plosone/s/licenses-and-copyright.

a) You may seek permission from the original copyright holder of Figures 2, 3, 4 and S2 to publish the content specifically under the CC BY 4.0 license. 

Reviewers' comments:

Reviewer's Responses to Questions

**Comments to the Author**

1. Is the manuscript technically sound, and do the data support the conclusions?

Reviewer #1: Yes

Reviewer #2: Yes

2. Has the statistical analysis been performed appropriately and rigorously? 

Reviewer #1: Yes

Reviewer #2: Yes

3. Have the authors made all data underlying the findings in their manuscript fully available?

Reviewer #1: Yes

Reviewer #2: Yes

4. Is the manuscript presented in an intelligible fashion and written in standard English?

Reviewer #1: No

Reviewer #2: Yes

5. Review Comments to the Author

Reviewer #1: This study indicates the possible association of the CboV-2 with encephalopathy in dogs. This research is of certain significance. But the paper needs to be thoroughly revised because of a number of spelling and grammar errors.

Reviewer #2: You have a nice and interesting manuscript here which has some very interesting findings in it which will be of interest to the wider scientific community. I have added some comments in below which hopefully are useful. They are mainly just typographical comments which I have tried to correct for you. But overall, a nice paper

Line 24- (CBoVs) have been recognised ….

Line 26- across is spelt incorrectly

Line 36- also detected in the intestine, lymphoid organs and the heart ….. (reword)

Line 40- CBoV-2

Line 47- belongs to the Parvoviridae family (reword)

Line 57- 2012 in dogs, has recently been considered to be pathogenic ….. (reword)

Line 86- except for one exception (reword)

Line 89- where ethe CBoV sites of replication (reword)

Line 114- buffered formalin manufacturer needed

Line 114- For the histopathology study … (reword)

Line 117- homogenised in what?

Line 120- nucleic acid

Line 120- quantified rather than quantitated may sound better?

Line 123- please define FFPE

Line 132- were taken from our previous publication (reword)

Line 133- please include the manufacturer and contents of the GoTaq Green Master mix

Line 136- the positively amplified fragments (reword)

Line 144- any reason why rabies wasn’t tested for?

Line 149- including the PCR reagents and conditions here would be useful

Line 155- coding sequences

Line 158- reference for MEGA 7

Line 166- reference or link for RDP may be useful

Line 170- manufacturer needed for Simplot

Line 173- how were the models chosen for the trees to be produced?

Line 176- used as a template … (reword)

Line 179- canine brains from this study…(reword)

Line 181- jModelTest reference needed

Line 194- reword this as unclear

Line 195- maybe selected rather than selective?

Line 199- 440bp of the NS1 gene .. (reword)

Line 221- was performed can be removed

Line 216- reword as unclear

Line 216-217- the selected two CBoV positive brain sections …(reword)

Line 217- virus particles in brain … (reword)

Line 235- just be careful with screening for genomes, you didn’t screen for a full genome, just a part of it. Maybe better to say PCR screening?

Line 240- why not included the sequence results here?

Line 241- Among the positive samples ….(reword)

Line 243- comma after (15/15)

Line 257- is this based on genome or genes- this needs to be made clear

Line 264- isn’t this complete genomes?

Line 271- comma after study

Line 271- is it really 2000 BC? Over 4000 years ago?

Line 272- you mention there being 6 before- why only 5 now?

Line 282- red circles represented CBoV strains collected in this study (reword)

Line 292- neutrophils (typo)

Line 302-303- what about the other samples?

Line 307- in the cerebellum and the brain stem (reword)

Line 308- using a feline parvovirus …. (reword)

Line 317- used as a counterstain (C) and (D). (reword and correct full stops)

Line 328- demonstration of an infected brain … (reword)

Line 330- delete ‘that’

Line 335- including the intestines (reword)

Line 334-336- this may be nice shown in a table

Line 337- is this genomes or genes?

Line 340- is this epithelia rather than epitheliums?

Line 354- bocaviruses (reword)

Line 361- definitively is spelled wrong

Line 381- genotypes (reword)

Line 383- epithelia rather than epitheliums

And line 385

Line 390- Our results also showed a tissue tropism of CBoV-2 in the myocardium (reword)

Line 392- is associated with myocarditis (reword)

Line 394- did you detect virus in the blood?

Line 397- also details viral gene/ genome detection (reword)

Line 398- other parts of the body (reword)

Line 399- may be caused by differing sensitivities (reword)

Line 403- has been investigated in several …. (reword)

Line 414- this would be useful to have a reference for

6. PLOS authors have the option to publish the peer review history of their article (what does this mean?). If published, this will include your full peer review and any attached files.

Reviewer #1: No

Reviewer #2: No

---

## [Author Response · Author response to Decision Letter 0]

7 Jul 2021

Reviewer's Responses to Questions

Comments to the Author

1. Is the manuscript technically sound, and do the data support the conclusions?

Reviewer #1: Yes

Reviewer #2: Yes

 Response: We appreciate with these reviewer comments. 

2. Has the statistical analysis been performed appropriately and rigorously? 

Reviewer #1: Yes

Reviewer #2: Yes

Response: We appreciate with these reviewer comments.

3. Have the authors made all data underlying the findings in their manuscript fully available?

Reviewer #1: Yes

Reviewer #2: Yes

Response: We appreciate with these reviewer comments.

4. Is the manuscript presented in an intelligible fashion and written in standard English?

Reviewer #1: No

Reviewer #2: Yes

 Response: We have submitted for proofreading by native English speaker and corrected the typo and grammatical errors. 

5. Review Comments to the Author

Reviewer #1: This study indicates the possible association of the CboV-2 with encephalopathy in dogs. This research is of certain significance. But the paper needs to be thoroughly revised because of a number of spelling and grammar errors.

 Response: We thank you for positive comments to our manuscript. As reviewer’s concern, we have submitted for proofreading by native English speaker and corrected the typo and grammatical errors.

 

Reviewer #2: You have a nice and interesting manuscript here which has some very interesting findings in it which will be of interest to the wider scientific community. I have added some comments in below which hopefully are useful. They are mainly just typographical comments which I have tried to correct for you. But overall, a nice paper

Response: We thank you for positive results to our manuscript and respect to reviewer’s comments and suggestions. We have revised as following suggestion as: 

Line 24- (CBoVs) have been recognised ….

Line 26- across is spelt incorrectly

Line 36- also detected in the intestine, lymphoid organs and the heart ….. (reword)

Line 40- CBoV-2

Line 47- belongs to the Parvoviridae family (reword)

Line 57- 2012 in dogs, has recently been considered to be pathogenic ….. (reword)

Line 86- except for one exception (reword)

Line 89- where ethe CBoV sites of replication (reword)

Response to Lines 24-89: We have revised and corrected the typos and grammatical errors as reviewer’s suggestion. 

Line 114- buffered formalin manufacturer needed

Response: We have added the manufacturer detail as “The collected brain samples were aliquoted for routine histology by immersing in 10% neutral buffered formalin (SigmaAldrich, MA, USA)” in page 5, lines 108-109.

Line 114- For the histopathology study … (reword)

Response: We have revised it as your suggestion. 

Line 117- homogenised in what?

Response: We have provided the details regarding tissue extraction as “Briefly, the brain tissue samples (5 g) were homogenized in 1X phosphate buffered saline using an aseptic technique.” in page 5, lines 112-113. 

Line 120- nucleic acid

Line 120- quantified rather than quantitated may sound better?

Response to Line 120: We have revised them as your suggestion. 

Line 123- please define FFPE

Response: We have defined the FFPE as described “Fresh-frozen and formalin-fixed, paraffin-embedded (FFPE) samples derived from various organs in the ED group were also collected.” in page 6, lines 118-119.

Line 132- were taken from our previous publication (reword)

Response: We have revised them as your suggestion. 

Line 133- please include the manufacturer and contents of the GoTaq Green Master mix

Response: We have included the manufacturer and contents of the GoTaq Green Master mix as described “Specifically, we used GoTaq Green Master Mix (Promega, Madison, WI, USA) which contains a mixture of Taq DNA polymerase, 400 µM of dNTPs, 3 mM of MgCl2 and PCR buffers, and added 5 µl of extracted nucleic acids to increase detection affinity.” in page 6, lines 128-130. 

Line 136- the positively amplified fragments (reword)

Response: We have revised them as your suggestion. 

Line 144- any reason why rabies wasn’t tested for?

Response: As laboratory safety, all the samples have tested for rabies infection using fluorescent antibody test prior inclusion to this study. As described in the Materials and methods section, we have excluded the rabies-positive brain samples from this investigation as described (page 5, line 101-102.). Additionally, we have revised the information in the Discussion section as “Since this study investigated on the rabies-negative brain samples, we only detected the CBoV-2 genome (with no presence of other common canine viral neuropathogens…” in page 16, line 360-362. 

Line 149- including the PCR reagents and conditions here would be useful

Response: We have provided the PCR reagents and conditions as described “The designed primer pair were used in a combination of GoTaq Green Master Mix (Promega, Madison, WI, USA) and 5 µl of DNA template, to amplify sequential CBoV fragments. The cycling conditions included 95°C, 5 min of initial denaturation, followed by 40 cycles of 95°C for 30 sec, 50°C for 1 min, 72°C for 12 min, with a final extension at 72°C for 10 min.” in page 7, lines 146-450.

Line 155- coding sequences

Response: We have revised them as your suggestion. 

Line 158- reference for MEGA 7

Line 166- reference or link for RDP may be useful

Line 170- manufacturer needed for Simplot

Response to Lines 158-170: We have added the reference regarding the MEGA 7 (page7, line 158), the RDP (page 8, line 166), and together with Simplot software (page 8, line 170).

Line 173- how were the models chosen for the trees to be produced?

Response: As default setting which is appropriated for the analysis of non-complex sequences as described in the references as we have added in the “The bootscan analysis was tested using the Kimura-2 parameter (K2P) model for 100 replications, with a window size of 200 bp, step size of 20 bp, and transition/transversion ratio of 2, as previously described [18, 27, 28, 34].” in page 8, lines 172-174.

Line 176- used as a template … (reword)

Line 179- canine brains from this study…(reword)

Response to Lines 176-179: We have revised as reviewer’ suggestion. 

Line 181- jModelTest reference needed

Response: We have provided the reference regarding the jModelTest as described in page 8, line 181.

Line 194- reword this as unclear

Response: We have rewritten the sentence as “To support the results of CBoV PCR and hence viral tropism in other organs, seven selected CBoV FFPE tissues in which CBoV PCR was positive were subjected to perform the ISH technique with a chromogenic DNA probe.” In page 9, lines 194-196.

Line 195- maybe selected rather than selective?

Line 199- 440bp of the NS1 gene .. (reword)

Line 221- was performed can be removed

Line 216- reword as unclear

Line 216-217- the selected two CBoV positive brain sections …(reword)

Line 217- virus particles in brain … (reword)

Response to Lines 195-217: We have reworded as reviewer’s suggestions. 

Line 235- just be careful with screening for genomes, you didn’t screen for a full genome, just a part of it. Maybe better to say PCR screening?

Response: We have corrected the word as described the subheading as “Genetic screening of CBoV in brain samples” (page 11, line 234) and the text as “As shown by the initial BoV screening results using a panBoV PCR assay in brain samples” in page 11, line 235. 

Line 240- why not included the sequence results here?

Response: Since the obtained sequences derived from the PBoV PCR assay (as screening assay) are only 141 bp in length and it was not possible to analyse them based on only short sequence. We, therefore, extended the sequencing results as described in the whole genome characterization part. 

Line 241- Among the positive samples ….(reword)

Line 243- comma after (15/15)

Response to Lines 241-243: We have reworded as reviewer’s suggestions. 

Line 257- is this based on genome or genes- this needs to be made clear

Response: We do confirm that the pairwise divergence was analysed based on the complete coding sequences as a result provided in the S1 Fig.

Line 264- isn’t this complete genomes?

Response: We analysed based on complete genome sequences. The GenBank accession nos. have provided within the Fig 1.

Line 271- comma after study

Response: We have revised as your suggestion. 

Line 271- is it really 2000 BC? Over 4000 years ago?

Response: We do apologize for this mistake. We have revised it as “We have reworded as reviewer’s suggestions.” in page 12, lines 270-272. 

Line 272- you mention there being 6 before- why only 5 now?

Response: We do apologize for uncleared information. We have revised it as “Based on reference strains, five of the six CBoV strains (036CP1, 038CP3, 008CP4, 240CP5, and 241CP6) belong to CBoV-2 GI, which shares the same lineage as an CBoV strain isolated from Hong Kong and South Korea and a BoV strain, 147CP7, belonging to CBoV-2 GII, which exhibits high nucleotide identity to a CBoV strain found in Thailand in 2016 (MG025952)” in pages 12-13, lines 272-276.

Line 282- red circles represented CBoV strains collected in this study (reword)

Line 292- neutrophils (typo)

Response to Lines 282-292: We have revised as reviewer’s suggestion. 

Line 302-303- what about the other samples?

Response: As we have included the 7 CBoV PCR-positive brain samples to perform the ISH and the results showed that 6/7 samples were positive to ISH assay, the remaining one samples showed negative ISH staining. Regarding this result, we already have discussed as “In the present study, although we detected CBoV-2 DNA in all the selected brain tissue samples, as well as in several other organs in the infected dogs, not all brain samples or PCR-positive tissues revealed ISH signals to support the presence of viral DNA. Hematogenous spreading of the virus or low amounts of virus present in the tested organs may be hypothesized for this finding.” in Discussion section (pages 17-18, lines 386-390.)

Line 307- in the cerebellum and the brain stem (reword)

Line 308- using a feline parvovirus …. (reword)

Line 317- used as a counterstain (C) and (D). (reword and correct full stops)

Line 328- demonstration of an infected brain … (reword)

Line 330- delete ‘that’

Line 335- including the intestines (reword)

Response to Lines 307-335: We have revised and corrected as reviewer’s suggestions. 

Line 334-336- this may be nice shown in a table

Response: We have already provided the result in S2 Table. 

Line 337- is this genomes or genes?

Response: We have revised as “To explore the potential distribution of CBoV-2 in different organs, both conventional PCR and ISH methods were used.” (Page 15, lines 331-332.)

Line 340- is this epithelia rather than epitheliums?

Line 354- bocaviruses (reword)

Line 361- definitively is spelled wrong

Line 381- genotypes (reword)

Line 383- epithelia rather than epitheliums

Response to Lines 340-383: We have revised as reviewer’s suggestions. 

Line 390- Our results also showed a tissue tropism of CBoV-2 in the myocardium (reword)

Line 392- is associated with myocarditis (reword)

Response to Lines 390-392: We revised them as reviewer’s suggestions.

Line 394- did you detect virus in the blood?

Response: We have not performed the CBoV PCR in the blood samples and we have discussed based on previous publications as described in page 18, lines 392-393. However, we have provided additional discussion as described “Hematogenous spreading of the virus or low amounts of virus present in the tested organs may be hypothesized for this finding. However, we did not investigate the presence of CBoV-2 in blood of these cases, further investigations are needed to support this hypothesis. Evidence of viral hematogenous spreading of CBoV-2 and other BoVs has been reported, in addition to viral gene/ genome detection in various organs [5, 18, 21, 22].” in page 18, lines 389-393.

Line 397- also details viral gene/ genome detection (reword)

Line 398- other parts of the body (reword)

Line 399- may be caused by differing sensitivities (reword)

Line 403- has been investigated in several …. (reword)

Response to Line 397-403: We have revised them according to reviewer’s suggestions. 

Line 414- this would be useful to have a reference for

Response: We have added the references regarding this as “This finding is supported by molecular epidemiology studies of CBoV strains isolated in different districts and countries [5, 10, 24, 43].” in pages 18-19, lines 410-411.

6. PLOS authors have the option to publish the peer review history of their article (what does this mean?). If published, this will include your full peer review and any attached files.

Do you want your identity to be public for this peer review? For information about this choice, including consent withdrawal, please see our Privacy Policy.

Reviewer #1: No

Reviewer #2: No

---

## [Decision Letter · Decision Letter 1]

16 Jul 2021

Canine Bocavirus-2 infection and its possible association with encephalopathy in domestic dogs

PONE-D-21-17173R1

Dear Dr. Piewbang,

We’re pleased to inform you that your manuscript has been judged scientifically suitable for publication and will be formally accepted for publication once it meets all outstanding technical requirements.

Kind regards,

Simon Clegg, PhD

Academic Editor

PLOS ONE

Additional Editor Comments:

Many thanks for resubmitting your manuscript to PLOS One

As you have addressed all the comments and the manuscript reads well, I have recommended it for publication

You should hear from the Editorial Office shortly.

It was a pleasure working with you and I wish you the best of luck for your future research

Hope you are keeping safe and well in these difficult times

Thanks

Simon

Reviewers' comments:

Reviewer's Responses to Questions

**Comments to the Author**

1. If the authors have adequately addressed your comments raised in a previous round of review and you feel that this manuscript is now acceptable for publication, you may indicate that here to bypass the “Comments to the Author” section, enter your conflict of interest statement in the “Confidential to Editor” section, and submit your "Accept" recommendation.

Reviewer #1: All comments have been addressed

Reviewer #2: All comments have been addressed

2. Is the manuscript technically sound, and do the data support the conclusions?

Reviewer #1: Yes

Reviewer #2: Yes

3. Has the statistical analysis been performed appropriately and rigorously? 

Reviewer #1: Yes

Reviewer #2: Yes

4. Have the authors made all data underlying the findings in their manuscript fully available?

Reviewer #1: Yes

Reviewer #2: Yes

5. Is the manuscript presented in an intelligible fashion and written in standard English?

Reviewer #1: Yes

Reviewer #2: Yes

6. Review Comments to the Author

Reviewer #1: This study indicates the possible association of the CboV-2 with encephalopathy in dogs. This research is of certain significance. The author has answered my previous questions and I think this article can be accepted.

Reviewer #2: I wish to thank the authors for their diligence in addressing my previous comments. I believe that the manuscript reads much better now.

There are three very minor comments which could be addressed during final editing as I dont feel that they warrant delaying the publication of this interesting manuscript.

With best wishes

Line 167- delete one of the s in settings

Line 225- from a previous study (reword)

Line 358- typo in definitively.

7. PLOS authors have the option to publish the peer review history of their article (what does this mean?). If published, this will include your full peer review and any attached files.

Reviewer #1: No

Reviewer #2: No

---

## [Editor Report · Acceptance letter]

4 Aug 2021

PONE-D-21-17173R1 

Canine bocavirus-2 infection and Its Possible Association with Encephalopathy in Domestic Dogs 

Dear Dr. Piewbang:

I'm pleased to inform you that your manuscript has been deemed suitable for publication in PLOS ONE. Congratulations! Your manuscript is now with our production department. 

Kind regards, 

on behalf of

Dr. Simon Clegg 

Academic Editor

PLOS ONE